# Adaptive Content Precaching Scheme Based on the Predictive Speed of Vehicles in Content-Centric Vehicular Networks

**DOI:** 10.3390/s21165376

**Published:** 2021-08-09

**Authors:** Youngju Nam, Hyunseok Choi, Yongje Shin, Euisin Lee, Eun-Kyu Lee

**Affiliations:** 1School of Information and Communication Engineering, Chungbuk National University, Cheongju 28644, Korea; imnyj@cbnu.ac.kr (Y.N.); plazpt@cbnu.ac.kr (H.C.); yjshin@cbnu.ac.kr (Y.S.); 2School of Information and Communication Engineering, Incheon National University, Incheon 21999, Korea

**Keywords:** content-centric vehicular networks, content precaching, vehicle mobility, predictive speed

## Abstract

Content-Centric Vehicular Networks (CCVNs) are considered as an attractive technology to efficiently distribute and share contents among vehicles in vehicular environments. Due to the large size of contents such as multimedia data, it might be difficult for a vehicle to download the whole of a content within the coverage of its current RoadSide Unit (RSU). To address this issue, many studies exploit mobility-based content precaching in the next RSU on the trajectory of the vehicle. To calculate the amount of the content precaching, they use a constant speed such as the current speed of the vehicle requesting the content or the average speed of vehicles in the next RSU. However, since they do not appropriately reflect the practical speed of the vehicle in the next RSU, they could incorrectly calculate the amount of the content precaching. Therefore, we propose an adaptive content precaching scheme (ACPS) that correctly estimates the predictive speed of a requester vehicle to reflect its practical speed and calculates the amount of the content precaching using its predictive speed. ACPS adjusts the predictive speed to the average speed starting from the current speed with the optimized adaptive value. To compensate for a subtle error between the predictive and the practical speeds, ACPS appropriately adds a guardband area to the precaching amount. Simulation results verify that ACPS achieves better performance than previous schemes with the current or the average speeds in terms of the content download delay and the backhaul traffic overhead.

## 1. Introduction

With the fast development in wireless communications and vehicular technologies, vehicular ad hoc networks (VANETs) have enabled us to deliver data to vehicles through wireless and mobile communication networks in vehicular environments [1]. Many projects (e.g., VIC’S [2], CarTALK 2000 [3], NOW (Network-on-Wheels)) and industry groups (e.g., the Car2Car Communication Consortium [4]) have conducted various research to provide the intelligent transport system by using VANETs. In the intelligent transport system, VANETs enable us to provide drivers and passengers with safety and convenience, and introduce new applications for entertainment and environment monitoring [5]. Many studies on VANETs have addressed various applications such as car accident warning for active safety, emergency vehicle access for public service, road congestion notice for improved driving, and commercial advertisement for business [6,7]. However, the existing VANETs protocols are considered as an inefficient approach to share vehicular contents between vehicles because they rely on a host-centric communication paradigm based on IDs.

The paradigm of Content-Centric Networking (CCN) has been proposed to provide a general communication form to achieve efficient content distribution and sharing on the Internet. CCN focuses on what (content) instead of where (host) [8]. In CCN, a content requester that is interested in an intended content broadcasts an interest packet for searching the content. Content publishers or providers having the content and receiving the Interest packet send out the content to the content requester via the reverse path of the interest packet. Then, nodes on the path of the content delivery cache the content in their own storage to instantly respond to interest packets from other content requesters [9]. Recently, a Content-Centric Vehicular Networking (CCVN) framework has been introduced to apply CCN in vehicular networks with RoadSide Units (RSUs) [10]. Instead of broadcasting an interest packet to search an intended content, the CCVN framework allows a requester vehicle to send the interest packet only to the RSU where it is located. The RSU delivers the interest packet to a content provider (i.e., a content server or a content publisher) of the content through its backhaul links, receives the content from the content server and downloads the content to the requester vehicle. Then, CCVNs also allow the RSU to cache the content to immediately respond to interest packets from other requester vehicles and thus enhance the performance of content downloading. Many studies on caching have been researched about where, what, and how long to cache contents on RSUs receiving them in CCVNs [11,12,13,14,15].

Owing to the development of various vehicular applications, both the number of contents and the number of requester vehicles increase rapidly [16]. Accordingly, a great number of demands for searching contents happens from a lot of vehicles in CCVNs. To efficiently handle the high demands, the research on proactive caching (i.e., precaching) have been actively conducted in CCVNs [17,18,19,20,21,22,23,24]. Precaching proactively caches anticipated contents for vehicles at RSUs from content servers in advance even though they have never been requested by vehicles. Many studies on precaching determine contents or RSUs for precaching by considering factors such as the mobility of vehicles or the popularity of contents. Since precaching enables vehicles to immediately download precached contents from RSUs without using content servers, it reduces the delay of content downloading and the overhead of backhaul link traffic in CCVNs. However, due to the large size of contents such as multimedia content (for example, videos on YouTube, Netflix, etc.), a vehicle has the difficulty to download the whole amount of content within the coverage of the RSU where it is currently connected [19]. As a result, when it enters the next RSU, it resends the interest packet to the next RSU. Thus, it should wait to download the content until the next RSU receives the content from the content provider. It increases the delay of content downloading.

To solve the problem of the content downloading delay for the large size of contents, the research on mobility-based precaching has been recently conducted by many studies in CCVNs [17,25,26,27]. The mobility-based precaching exploits precaching of contents in continuous next RSUs on the trajectory of the vehicle. In this precaching method, an intended content requested for a vehicle is proactively cached to the next RSU on its trajectory before it arrives at the next RSU. For content precaching, these schemes proactively cache the same copy of the whole content (i.e., full precaching) for the vehicle to the next RSU where it arrives after leaving the current RSU [11]. However, since the full precaching caches the unnecessary parts (that the vehicle already downloaded in the current RSU or cannot download in the next RSU) of the content to the next RSU, it increases traffic of backhaul links and overheads of caching storages. To solve this problem of the full precaching, several schemes have been proposed to proactively cache only the downloadable amount (i.e., partial precaching) of the content for the vehicle in the next RSU [28,29,30,31]. In the partial precaching, the information of vehicle speeds is used to calculate the downloadable amount of the content. Then, these schemes use only the constant speed as the vehicle speed information which is the current speed [28,29,31] of the requester vehicle or the average speed [30] of vehicles in the next RSU. However, since the current speed and the average speed do not appropriately reflect the practical speed of the requester vehicle in the next RSU, the previous schemes could incorrectly calculate the downloadable amount of the content in the next RSU. If the downloadable amount of the content is calculated more than the amount that the vehicle can practically download in the next RSU, it increases the overhead of backhaul traffic [32]. In contrast, if the downloadable amount of the content is calculated less than the amount that the vehicle can practically download, it increases the delay of content downloading.

Therefore, we propose an adaptive content precaching scheme (ACPS) that efficiently reduces the delay of content downloading and the overhead of backhaul traffic for the mobility-based partial precaching in CCVNs. First of all, ACPC estimates the predictive speed of a requester vehicle in the next RSU to accurately reflect its practical speed. The predictive speed of the requester vehicle is adjusted to the average speed of vehicles in the next RSU from starting with the current speed of the requester vehicle by the optimized adaptive value. Then, ACPS calculates the downloadable amount of the content for the requester vehicle in the next RSU from its predictive speed. Since the predictive and the practical speeds of the requester vehicle in the next RSU might have a subtle error between them, the hit ratio of precaching about the downloadable amount of the content is reduced and thus the content downloading delay raises. ACPS adds a guardband area to the downloadable amount to compensate for the subtle error of the speed prediction. Then, ACPS carries out precaching of the downloadable amount with the guardband area in the next RSU. As a result, the requester vehicle can immediately download the precached content from the next RSU when it arrives at the next RSU. Simulation results conducted in various environments verify that ACPS achieves better performance than a current speed scheme [31] and an average speed scheme [30] among the previous mobility-based partial precaching schemes in terms of the content downloading delay and the backhaul traffic overhead.

Our goal in this paper is to correctly calculate the downloadable amount of an intended content to efficiently precache it in the next RSU in terms of the content downloading delay and the backhaul traffic overhead. To achieve this goal, our contribution can be summarized as follows.

To calculate the precached amount of an intended content in the next RSU, ACPS estimates the predictive speed of a requester vehicle by considering both the current speed of the requester vehicle and the average speed of vehicles in the next RSU and applying the optimized adaptive value, and thus reduce the backhaul traffic overhead by preventing unnecessary precaching.To compensate for the difference between the predictive and the practical speeds of the requester vehicle in the next RSU, ACPS includes a guardband added in the precached amount of an intended content to completely download the intended content, and thus reduces the content downloading delay by raising the hit ratio of precaching.To evaluate the performance of the proposed scheme, ACPS is compared with two previous schemes, the current speed scheme and the average speed scheme in simulations of NS3 with including the Routes Mobility Model extracted with API key from Google Map Platform. ACPS has better performance than the previous scheme for precaching in CCVNs.

The remainder of this paper is organized as follows. We examine the related works of the proposed scheme in Section 2. The network model and problem statement for the proposed scheme is presented in Section 3. The proposed scheme is described in detail in Section 4. The performance evaluation of the proposed scheme is verified in Section 5 through simulation results. Section 6 concludes this paper.

## 2. Related Works

### 2.1. Caching in CCVNs

In Content-Centric Vehicular Networks (CCVNs) [10,33], since the capabilities (e.g., storage, battery, sensor, memory, etc.) of vehicles and road-side infrastructures (e.g., Access Points (APs), Road-Side Units (RSUs), local servers, etc.) are improved, they enable us to cache the large size contents [34,35]. Thus, several studies [11,12,13,14,15] have been researched about where, what, and how long to cache the contents in CCVNs. Blaszczysyzn et al. [11] proposed the optimal probabilistic placement policy of content caching, which guarantees the maximal hit probability of the content caching in random network topologies. To achieve the maximal hit probability, it exploits multi-coverage regions and delivers considerable performance improvement compared to the standard *cache the most popular content, everywhere* scheme. It formulates and solves the problem of an optimal randomized content placement policy to maximize the user’s caching hit probability. Ostrovskaya et al. [12] proposed a new multi-metric content replacement policy (M2CRP) for content stores in NDN-driven VANETs. M2CRP considers three metrics to collectively encompass the requirements for the improved performance of VANETs applications: the freshness of the content, the popularity, and the distance between the locations where the content was received and saved in content stores and the current location of the caching node. Marica et al. [13] proposed a novel caching strategy, referred to as Diversity improved cAchiNg of popular Transient contEnts (DANTE) where vehicles autonomously decide which content is to be locally cached according to the content residual lifetime, its popularity and the perceived availability of the same content in the neighborhood. They also design a set of minor modifications in the Named Data Networking (NDN) node architecture and packet fields to support DANTE operations. Caching decisions are taken by vehicles autonomously. Vehicles can find the majority of distinct fresh and popular contents nearby, without flooding the network with content requests that have to reach the original source. Deng et al. [14] proposed a Distributed Probabilistic Caching (DPC) scheme to solve the problems such as high content retrieval latency, low cache space utilization, and high content redundancy caused by the always cache policy in VANETs through NDN. DPC exploits three main factors to decide caching of contents: the demand and preference of vehicles, the importance of vehicles in the network, and relative movement of the receiver and the sender. Dua et al. [15] proposed a content caching scheme based on a bloom filter model (that is the probability data structure) to improve efficient content distribution in CCVNs. The bloom filter model is used to achieve better time complexity and quicker content insertion, deletion and search processes. Through the bloom filter model, the scheme allows vehicles to have the functionality of cache to support cooperative content distribution between them.

### 2.2. Precaching in CCVNs

In CCVNs for VANETs, unlike the traditional CCNs for the Internet, a requester vehicle cannot stay in an RSU for a long time to completely download its intended contents from the RSU because it passes the RSU due to its high mobility on roads in vehicular networks. Thus, only a single RSU may not be able to provide the whole of a content to a vehicle connected to it within the short time in its communication coverage. The requester vehicle may need to pass the coverages of multiple RSUs to download the whole content. Therefore, some studies [17,18,19,20,21,22,23,24] have been researched to precache the content before a requester vehicle enters into the coverage of the next RSU. Kanai et al. [17] proposed a proactive content caching scheme that utilizes transportation systems to provide high quality and highly reliable video delivery through efficient wireless resource usage. The requested content is divided into several segments which are delivered separately to relay points according to the traveling schedule of transportation vehicles such as trains or buses. The transportation vehicle receives the distributed content segments at relay points. The transportation vehicle streams the received content segments to mobile users inside it via wireless LANs. Yao et al. [18] proposed a scheme called cooperative caching based on mobility prediction (CCMP) for CCVNs. The main idea of CCMP is to cache popular contents at a set of mobile nodes that may visit the same hot spot areas repeatedly. CCMP uses a prediction based on partial matching to predict mobile nodes’ probability of reaching different hot spot regions based on their past trajectories. Zhou et al. [19] proposed a novel content delivery framework by leveraging the 5G edge networks in which the content caching and data prefetching techniques are exploited to provide vehicular content distribution. They design a system architecture leveraging RSUs named vehicular edge infostations for edge vehicular content delivery in 5G networks. To reduce the data access delay and efficiently utilize the precious connection time of vehicles, the infostations are featured by the deployment of local content servers and the use of content caching and data prefetching techniques. Luo et al. [20] proposed EdgeVCD, an intelligent algorithm-inspired content distribution scheme. Specifically, they first propose a dual-importance (DI) evaluation approach to reflect the relationship between the Priority of Vehicles (PoV) and the Priority of Contents (PoC). To solve the complex optimization problem effectively, they first divide the road into small segments. Then, they propose a fuzzy logic-based method to select the most proper content replica vehicle (CRV) for supporting content distribution efficiently and redefine the number of content requester vehicles in each segment. Hui et al. [21] proposed a zone-based content precaching strategy, which aims to implement an active content caching through precaching zone selecting algorithm and precaching node selecting algorithm. They organize the edge servers (ESs) with a zone-based way at the edge, and assign a Manager node to collect the information of each zone. A content precaching zone can be selected by comparing estimated request delay and zone sojourn time. To find an ES with a lower workload to reduce the future response delay, ES can be selected by its centrality, load degree and content popularity. Lin et al. [22] proposed a Cooperative Caching scheme based on Social Attributes and Mobility Prediction (CCSAMP) for VCCN. It is based on the observation that vehicles move around and are liable to contact each other according to drivers’ common interests or social similarities. They incorporate both social attributes and hot-zone visiting probability into the design of a cooperative caching scheme for VCCN. They also propose a cache replacement policy that evaluates the content popularity by combining the social similarity and time interval of two consecutive requests. Ahmed et al. [23] proposed a system model that makes use of the cached contents on passing vehicles in order to fill up the RSU cache such that the RSU can later serve the requests arising from vehicles without need to access the backhaul link. They define the system as Markov Decision Process (MDP) and specify its items (state, action, reward) and use a DQN-learning and design a heuristic algorithm to solve the caching problem. The DQN-based solution does not stick with certain contents, it learns the best policies which optimize the performance based on the utility of the cached contents and their size. Ruyan et al. [24] proposed a cooperative caching strategy with content request prediction (CCCRP) in IoV, which precaches the contents requested by vehicles with greater probability in other vehicles or the RSU to reduce the content acquisition delay. They use the Long Short Time Memory (LSTM) model to predict the number of content requests on the time series with different the processing mode. In addition, they mainly focus on the caching decision and propose a reinforcement learning-based algorithm for the cooperative caching strategy with content request prediction (CCCRP) in IoV according to the prediction results of content requests. Table 1 shows the summary of the related works on precaching in CCVNs.

### 2.3. Mobility-Based Precaching in CCVNs

Since the above-mentioned precaching schemes do not consider the mobility information of a requester vehicle, the requester vehicle should search the RSUs which have the precached content to download the whole content. It causes reducing the cache hit ratio and the content delivery ratio. To efficiently download the whole content and to enhance the cache hit ratio, some studies [25,26,36,37,38,39] on mobility-based precaching that exploit the mobility of the requester vehicle have been researched in CCVNs. They proactively cache the same copy of the whole content (i.e., full precaching) for the requester vehicle to the next RSU where it arrives next by its mobility. Abani et al. [26] proposed a proactive caching scheme that leverages the flexibility of ICNs in precaching contents anywhere in the networks. They used entropy to measure the uncertainty of Markov-based mobility predictions to locate the next location of a requester vehicle and to make a strategic decision of where to precache in the networks. The uncertainty measurements determine the best precaching RSU around the mobility of the requester vehicle and thus eliminating redundant precaching. Zhao et al. [25] proposed a vehicle mobility prediction-assisted OTT prefetching mechanism in VANETs. They suggest a hierarchical VANETs architecture that enables vertical aggregation to optimize the precaching operations and improves system performances. They implemented a vehicle mobility prediction module to estimate the future connected RSUs using the dynamic Markov chain model and data traces collected from a real-world VANETs test-bed deployed in the city of Porto, Portugal. The mechanism increases the percentage of offloaded traffic, reduces the backhaul data volume, and optimizes the content retrieval latency. Sara et al. [36] proposed a scheme called Proactive Caching at Parked Vehicles (PCPV) to provide a better quality of service to vehicular users who tend to have a somewhat consistent social networking behavior. Users have a predictive behavior in terms of the type and time of access of social media platforms as a part of their daily routine during transit from one place to another. This process is done using a heuristic greedy approach to precache the data at the proper time and place based on future requests, trajectories, and estimated period of encounter with road segments. Zhang et al. [37] proposed RapidVFetch to facilitate vehicular data downloading by precaching data via V2V communication over NDN. Since it is the vehicle who best knows its future locations and desired content, RapidVFetch lets each vehicle, called a requester, express their needs by Interest packets which are small in size and solicit help from other vehicles, called forwarding vehicles, through V2V communication. Vehicles at future locations can simply express these small Interest packets to the RSUs, where the prefetched packets will be cached at the RSUs and will soon be used by the requester when it moves into the RSU’s range. Din et al. [38] proposed a Left-Right-Front (LRF) cache strategy for precaching in VANETs. The LRF cache strategy proactively places the requested data at upcoming nodes/RSUs for vehicles. This strategy works with a predefined ICN architecture without making a change in the existing data structures to cope with the problems of the dynamic nature of network and mobility due to IoT-based VANET nodes in the ICN environment. This strategy is suitable for improving cache utilization, hop ratios, and resolved interest ratios. Hu et al. [39] proposed a Peer-to-Peer Federated learning-based proactive Caching scheme (PPFC) that is well suited to the highly dynamic IoV environments. Due to the heterogeneous abilities of vehicles, a dual-weighted model aggregation scheme is designed to reduce the effect of straggler vehicles in order to further improve the accuracy of the trained global model in the designed peer-to-peer FL. PPFC can eliminate the issue of hand-over between RSUs, achieve lower latency and adapt to the mobility of vehicles. PPFC utilizes a Collaborative Filtering based Variational AutoEncoder (CF-VAE) model to predict content popularity based on the contextual information of users for making smart precaching decisions.

However, since the full precaching schemes [25,26,36,37,38,39] precache the whole of an intended content for a requester vehicle to its next RSU, they have the problem to precache unnecessary parts of the content that the vehicle already downloaded in the current RSU or cannot download in the next RSU, because the vehicle can download only the limited amount in the next RSU due to its limited travel time in the RSU. Thus, many studies [27,28,29,30,31,40,41] have been addressed to solve this problem and proposed partial precaching schemes to precache only the amount that a requester vehicle can download in the communication coverage of the next RSU through the communication with it based on the speed information. To calculate the amount of the precached content, these schemes use the constant speed as the vehicle speed information such as the current speed [27,28,29,31,40,41] of the requester vehicle or the average speed [30] of vehicles in the next RSU. Grewe et al. [28] proposed a novel proactive caching scheme for VANETs based on NDN. It calculates a list of chunks used to transmit an intended content of a requester vehicle based on the content size and the maximum payload. Then, it determines the next RSU on the trajectory of the requester vehicle for precaching chunks. The determination is based on the vehicle’s position, its velocity, and its INTEREST frequency. The number of total chunks that the requester vehicle is possible to download is calculated by the current constant speed vector of the vehicle. Guo et al. [27] proposed a novel cooperative communication scheme in consideration of both content prefetching and carry-and-forward methods to reduce the dark area between two neighbor RSUs and indirectly extend the coverage of an RSU for downloading the large-size contents. In the communication range of the RSU, a requester vehicle requests to download an intended content from a content server and requests the next two arrival RSUs to precache an appropriate portion of the remaining content for downloading the intended content. The expected downloadable amount for precaching in the next RSU is calculated by its transmission rate and the sojourn time of the requester vehicle in it. The sojourn time is calculated by the current speed of the requester vehicle and the communication coverage distance of the RSU. Khelifi et al. [29] proposed an optimized precaching scheme called PCMP for VANETs on the top of the NDN architecture, which predicts the next arrival RSU based on the LSTM module and precaches the intended content on the RSU. They calculated the number of chunks using the current velocity of the requester vehicle and the distance of the path at the coverage of RSU. Then, PCMP divides the intended content into chunks, then calculates the number of chunks required to be precached and downloaded in the next RSU. For the calculation, it exploits the connectivity duration of the requester vehicle within the coverage of the RSU and the link bandwidth between the vehicle and the RSU. Lin et al. [40] proposed a mechanism for precaching chunks of large content objects such as videos among RSUs. They adopt the Hidden Markov Model (HMM) to predict the moving trajectory of the requester vehicle. Based on the time gap from the current location to the predicted location of the requester vehicle, they calculate the size of video chunks for precaching by using the current driving speed of the requester vehicle. Then, the corresponding RSU can precache the required video chunks and provide them to the requester vehicle as soon as it arrives at the predicted location. Zhe et al. [31] proposed a novel hierarchical proactive caching approach that considers both the future demands of autonomous vehicle users and their mobility. This approach uses the non-negative matrix factorization (NMF) technique to predict user’s preferences which are then used to predict users’ future demands by considering the historical popularity of videos. They calculate the number of video chunks for precaching by using the arrival and departure time of the user at an edge node based on its current velocity vector. They consider not only the predicted ratings, but also the previous popularity of videos to predict the users’ future demands. Park et al. [41] proposed a mobility-aware distributed proactive caching scheme in CCVNs. To reduce the redundancy and the burden of precaching for multiple candidates next RSUs from the current RSU, the proposed scheme distributes the whole of the intended content to each of them as much as the mobility probability of the requester vehicle about each candidate next RSU, which means the probability to move to the candidate next RSU from the current RSU by the requester vehicle based on the Markov Model. They calculate the maximum number of chunks using the constant vehicle speed, and these chunks are distributively precached to each of the candidate next RSUs which the requester vehicle may travel. Zhang et al. [30] proposed a content caching and prefetching framework where an AP has separately a cache buffer and a prefetch buffer for popularity-based content caching and mobility prediction-based prefetching. For precaching to buffer, they calculate what chunks the client will request during the average residence time (that is calculated by the average speed of vehicles) based on the residence time history table. The framework can capture both long-term aggregated content access pattern and short-term individual user access pattern, thus considerably improving the cache hit ratio. They developed a network-level mobility prediction model to determine the next AP in the MobilityFirst architecture, which takes into consideration the latest mobility information from nearby mobile devices. As a result, since the current speed and the average speed does not appropriately reflect the practical speed of the requester vehicle in the next RSU, these schemes could incorrectly calculate the amount of the precached content. However, since ACPS uses the predictive speed of the requester vehicle that can adequately reflect its practical speed, it could calculate the amount of the precached content more accurately than them. Table 2 shows the summary of the related works on mobility-based precaching in CCVNs.

## 3. Network Model and Problem Statement

### 3.1. Network Model

As the model of a vehicular network, we consider roads where a great number of vehicle moves and a large number of RSU are deployed. In this network, every vehicle moves along its travel route to arrive at its destination by passing several RSUs. Then, it periodically sends beacon messages with its ID, current speed, current location, travel trajectory, and so on to each RSU to set up the communication to download contents [42]. In the proposed scheme, the interval of beacon massages is 0.1 s as the vehicular communication standard. When a vehicle wants to download an intended content, it sends a request message (called as an Interest packet in CCN) for the content to the RSU where it currently connects [10,19]. We name this vehicle as a requester vehicle to distinguish it from other vehicles. The request message includes the requester vehicle’s information such as its ID, position, current speed, travel route and type of content. When receiving the request message, the RSU delivers the request message to the closest content server on the Internet through backhaul networks [25]. If the intended content cannot be totally downloaded to the requester vehicle from the RSU due to its large size, the content server conducts precaching of the content to the next RSU where the requester vehicle will arrive next on its travel route.

In this paper, we also consider a Markov prediction model of the second-order to predict the trajectory of a vehicle and to determine its next RSU [26,43]. Usually, the first-order Markov model constructs a set of states (representing an RSU) $L=L_{1},L_{2},\ldots ,L_{n}$ and transition probabilities $p_{ij}$, which is the probability that the vehicle will be next connected to the RSU in $L_{j}$ when it is currently connected to the RSU in $L_{i}$. This is the property of the first-order Markov model in which the following states depend only on the current state. Then, the transition probability is defined as:(1)$$p_{ij}=Pr\left(L_{j}|L_{i}\right)=\frac{X(L_{i},L_{j})}{Z(L_{i})}$$
where $X(L_{i},L_{j})$ is the number of vehicles moved from the RSU in $L_{i}$ to the RSU in $L_{j}$, and $Z(L_{i})$ is the total number of vehicles that moved through the RSU in $L_{i}$. Then, second-order Markov model has the transition probabilities $p_{ik,j}$, which is the probability that the vehicle will be next connected to the RSU in $L_{j}$ when it is currently connected to the RSU in $L_{i}$ and was previously connected to the RSU in $L_{k}$. The transition probability $p_{ik,j}$ is defined as:(2)$$p_{ik,j}=Pr\left(L_{j}|L_{i},L_{k}\right)=\frac{X(L_{k},L_{i},L_{j})}{Z(L_{i})}$$
where $X(L_{k},L_{i},L_{j})$ is the number of vehicles moved from the RSU in $L_{k}$ to the RSU in $L_{i}$ via the RSU in $L_{j}$. Based on the Routes Mobility Model [44], we use the Brooklyn Taxi Movement dataset, which contains coordinates of approximately 100 taxis collected over a week in the Brooklyn area. We use the dataset to build the Markov model. As a result, the Markov model determines the next RSU of the requester vehicle.

### 3.2. Problem Statement

If the next RSU is determined, the content server calculates the amount of the content that the next RSU can provide to the requester vehicle within its communication coverage by precaching. To calculate the amount of the precached content, the existing precaching schemes use the transmission rate of the RSU and the travel time of the requester vehicle within the coverage of the RSU [27,28,29,31]. To expect the travel time, the existing schemes exploit the current speed ($V_{Cur}$) [27,28,29,31] of the requester vehicle or the average speed ($V_{Avg}$) [30] of vehicles within the next RSU. Then, they calculate the amount of the precached content by using the travel time. However, since both the current speed and the average speed are unchanging values as shown in Figure 1, they do not reflect the practical speed $V_{Practical}\left(w\right)$ (that is changeable) of the requester vehicle in the next RSU. Thus, in the existing schemes, the requester vehicle leaves the communication range of the next RSU earlier or later than its expected travel time. As a result, since they use the wrong travel time, they cause the increment of the content download delay and the backhaul traffic overhead.

Therefore, to solve this problem effectively, we use the predictive speed $V_{Predictive}\left(w\right)$ of the requester vehicle different from the existing schemes in order to accurately expect the travel time in the next RSU. As shown in Figure 1, the predictive speed is adjusted from the current speed to the average speed to properly reflect the practical speed. Let *t* be the travel time of the requester vehicle in the next RSU. It can be calculated by the following Equation (Equation 3):(3)$$t=\int_{in}^{out}\frac{1}{V(w)}dw$$
where in and out are locations of the entrance and the exit in the next RSU, respectively, and V(w) is the speed of the vehicle at the location *w* in the next RSU. Generally, since *t* can be the available content downloading time from the next RSU, it is used to calculate the amount of the precached content. Thus, predicting V(w) precisely is one of the very important issues to get *t*. However, it is very difficult and complex to accurately predict the practical speed $V_{Practical}\left(w\right)$ of the requester vehicle in the next RSU. Fortunately, vehicles have a property that their own practical speed $V_{Practical}\left(w\right)$ in an RSU might be generally converged to the average speed of vehicles in the RSU due to the speed limitation in the urban environment. Using this property, we calculate the predictive speed of the requester vehicle in the next RSU, which can be similarly matched to its practical speed. For providing the low complexity of the speed prediction calculation, we use the average speed, the current speed, and the adaptive value *a* as input values. Then, we prove the following objective Equation (Equation 4) and derive its results.
(4)$$\begin{array}{cc} \; & |\int_{in}^{out}V_{Practical}{\left(w\right)}^{-1}dx-\int_{in}^{out}{V_{Cur}}^{-1}dx|\\ \; & \geq |\int_{in}^{out}V_{Practical}{\left(w\right)}^{-1}dx-\int_{in}^{out}{V_{Avg}}^{-1}dx|\\ \; & \geq |\int_{in}^{out}V_{Practical}{\left(w\right)}^{-1}dx-\int_{in}^{out}V_{Predictive}{\left(w\right)}^{-1}dx| \end{array}$$

Nevertheless, the predictive speed of the requester vehicle may be different from its practical speed due to various road conditions (e.g., vehicular accident, traffic jam, road building, etc.). The difference (that is, the prediction error) between the predictive speed and the practical speed affects the downloading amount of the precached content. Due to this difference, the requester vehicle may not download efficiently the amount of the precached content in the next RSU. To solve this problem, we use a guardband to compensate for the prediction error, which is the additional amount of the precached content. Although the guardband raises the amount of the precached content, it enhances the performance of the content download. In the next section, we present the proposed scheme to solve the addressed problems in detail.

## 4. The Proposed Scheme

The proposed scheme uses precaching of a content in the next RSU on the trajectory of a requester vehicle. The amount of the precached content in the next RSU is determined by the predictive speed of the requester vehicle in the next RSU. The predictive speed is calculated by considering both the average speed of vehicles in the next RSU and the current speed of the requester vehicle. Since we mention several speeds in this paper, their definitions are as follows to clearly explain the proposed scheme.

Current speed ($V_{Cur}$): is defined as the speed of the requester vehicle at the time when it requests an intended content by sending an interest packet to the current RSU. It depends on situations of current traffic in the coverage of the RSU. It is used to calculate the predictive speed of the requester vehicle in the next RSU.Average speed ($V_{Avg}$): is defined as the speed that all vehicles averagely move within the coverage of the next RSU. With the information included in beacon messages, it is determined from collecting the historical data about the speeds of vehicles passed through the next RSU. The collected historical data are measured according to the time of the day. The average speed is continuously considered and managed at hourly intervals (for example, 1 a.m. to 2 a.m., 2 a.m. to 3 a.m., …, 11 p.m. to 12 p.m., and 12 p.m. to 1 a.m.) for weekdays and weekends. It is also used to calculate the predictive speed of the requester vehicle in the next RSU.Predictive speed ($V_{Predictive}\left(w\right)$): is defined as the speed that the requester vehicle is predicted to move within the next RSU by our scheme. It is calculated by adjusting to the average speed starting with the current speed through an acceleration factor.Practical speed ($V_{Practical}\left(w\right)$): is defined as the speed that the requester vehicle actually moves within the next RSU. It may be different from the predictive speed and the difference is the error of the speed prediction in our scheme.

To determine the amount of the precached content, the proposed scheme additionally needs two values of information about the requester vehicle in the next RSU as described in Figure 2. The first one is the location that means the coverage of the RSU from the point *in* to the point *out* and is symbolized as *w*. The second one is the communication rate for the distance between the RSU and the location *w* of the requester vehicle and is symbolized as *r${}_{w}$*.

In the following subsections, we describe in detail how to adjust the amount of the precached content with the predictive speed of the requester vehicle according to the cases of the correlation between *V${}_{Cur}$* and *V${}_{Avg}$*. In Section 4.1 and Section 4.2, we address the case of *V*${}_{Cur}$< *V*${}_{Avg}$ and the case of *V${}_{Cur}$ > V${}_{Avg}$*, respectively. Since the speed of the requester vehicle can be changed at any time, the predictive speed and the practical speed of the requester vehicle may be different. Based on the predictive speed of the requester vehicle, it may not fully download the amount of the precached content. Section 4.3 describes the addition of a guardband to the amount of the precached content for the requester vehicle to efficiently download the whole amount of the content.

When a requester vehicle enters the coverage of an RSU, its current speed *V${}_{Cur}$* can be lower than the average speed *V${}_{Avg}$* of vehicles passed through the RSU. This case might happen due to the increment of traffic different from the general circumstance of traffic on the road before the RSU. Since the speed of the requester vehicle is inconstant and the coverage of the RSU is able to averagely provide *V${}_{Avg}$*, the requester vehicle might increase its speed to *V${}_{Avg}$*. However, if it is assumed that the requester vehicle travels with only *V${}_{Cur}$* within the coverage of the RSU, the requester vehicle should request to download the additional amount of the content from content providers through backhaul links. Because, there is no more precached amount of the content after the requester vehicle fully downloads the amount of the precached content before it leaves the coverage of the RSU. Thus, we consider this feature on the changeable speed of the requester vehicle in the coverage of the RSU. The proposed scheme calculates the amount of the precached content in the RSU by considering the changeable speed.

To help better understand the calculation of the precached amount, we first explain the condition that the requester vehicle has a constant speed. If the speed of the requester vehicle is constant (that is, non-changeable), the precached amount ($C_{N,i}$) of the content that can be downloaded on the *i*th RSU without correction of the content size is calculated as shown in Equation (Equation 5),
(5)$$C_{N,i}=\int_{in}^{out}\frac{r_{w}}{V_{Cur}}dw$$
where *in* and *out* mean locations that the requester vehicle enters and leaves the coverage of the RSU, respectively. The amount of the precached content calculated by Equation (Equation 5) is shown as (a) in Figure 3 and Figure 4.

### 4.1. The Case of $V_{Cur}<V_{Avg}$

We define two values, *a* and *m* to consider the changeable speed of the requester vehicle in order to calculate the amount of the precached content. The value *a* is defined as an acceleration factor. It is the value that the requester vehicle accelerates to increase its speed from the current speed to the average speed, and is used as an input parameter in the simulation. The value *m* is defined as the location of the requester vehicle in the coverage of the RSU at the time that the current speed is equal to the average speed. Generally, the requester vehicle moves with *V${}_{Cur}$* at the point *in* in the coverage of the RSU. By moving with *a*, its speed increases and next is equal to *V${}_{Avg}$* at the location *m*. It continuously moves with *V${}_{Avg}$* after the location *m* and eventually, it leaves the location *out* in the coverage of the RSU. In this situation, as *a* increases, *m* is closer to the point *in*. In this case, the amount of the precached content calculated with the changeable speed is closer to the amount of the precached content calculated with *V${}_{Avg}$*. On the other hand, when decreasing *a*, *m* is farther from the point *in* and is closer to the point *out*. In this case, the amount of the precached content calculated with the changeable speed is closer to the amount of the precached content calculated with *V${}_{Cur}$*.

**Theorem** **1.**
*The amount of the precached content calculated with the changeable speed is dependent on the location m. m is calculated from Equation (Equation 6).*
(6)$$m=\frac{V_{Avg}-V_{Cur}}{a}\times \left(V_{Avg}-\frac{V_{Avg}-V_{Cur}}{2}\right)$$


**Proof.** See Appendix A. □

Thus, as aforementioned, the amount ($C_{C,i}$) of the precached content depends on the changeable speed of the requester vehicle in the coverage of the RSU. We define the changeable speed as *V${}_{w}$*. The amount ($C_{C,i}$) of the precached content based on *V${}_{w}$* is calculated by Equation (Equation 7) and shown in (a) of Figure 3.
(7)$$C_{C,i}=\int_{in}^{out}\frac{r_{w}}{V_{w}\left(w\right)}dw$$

In Equation (Equation 7), *V${}_{w}$* means the changeable speed of the requester vehicle when increasing the changeable speed of the requester vehicle from *V${}_{Cur}$* to *V${}_{Avg}$* until reaching to the point *out* by the requester vehicle.

**Theorem** **2.**
*V${}_{w}$ is calculated as shown in Equation (Equation 8),*
(8)$$\begin{array}{cc} V_{w}\left(x\right) & =\sqrt{{V_{Cur}}^{2}+2ax}\;if\left(m<out,x<m\right)or\left(m>out\right)\\ \; & =V_{Avg}\;\;\;\;if\left(m<out,x>m\right) \end{array}$$


**Proof.** See Appendix B. □

### 4.2. The Case of $V_{Cur}>V_{Avg}$

In this case, a requester vehicle may decrease its speed. However, the requester vehicle cannot fully download the amount of the precached content because the current speed of the requester vehicle is faster than the average speed. When the speed change of the requester vehicle is not considered, the amount of the precached content calculated by Equation (Equation 7) is shown as (c) in Figure 4.

**Theorem** **3.**
*Since V${}_{Cur}$ is larger in the case of $V_{Cur}>V_{Avg}$, $V_{w}\left(x\right)$ is transformed into Equation (Equation 9).*
(9)$$\begin{array}{cc} V_{w}\left(x\right) & =\sqrt{{V_{Cur}}^{2}-2ax}\;if\left(m<out,x<m\right)or\left(m>out\right)\\ \; & =V_{Avg}\;\;\;if\left(m<out,x>m\right) \end{array}$$


**Proof.** See Appendix C. □

### 4.3. Addition of Guardband

As mentioned above, the proposed scheme adjusts the speed of a requester vehicle. However, since the speed of the requester vehicle can be changed at any time, it is difficult to calculate the difference between the predictive speed and the practical speed of the requester vehicle. Thus, it is difficult to calculate the amount of the precached content precisely. As a result, the requester vehicle may not fully download the amount of the precached content. Thus, to address this issue, we add a guardband (i.e., an extra amount of the content) to the amount of the precached content calculated by the proposed scheme for the requester vehicle to fully download the amount of the precached content.

When a guardband is added to increase the precaching hit ratio about the amount of the precached content for the requester vehicle, Equation (Equation 10) is derived from Equation (Equation 5) by applying the guardband and is shown in (b) of Figure 3 and Figure 4.
(10)$$C_{NG,i}=\frac{\left(100+G\right)}{100}\times \int_{in}^{out}\frac{r_{w}}{V_{Cur}}dw$$

In Equation (Equation 10), *G* is a constant value between 0 and 100, and its optimal value is derived through experiments. If *G* is 0, no guardband is used. If *G* is 100, the precached content is doubled in size. In the case of $V_{Cur}<V_{Avg}$, the precached content is larger than the average amount of the precached content in (c) of Figure 3. Therefore, the point *m* where the speed of the requester vehicle is equal to the average speed in the RSU must be adjusted. To do this, we derive the amount of the precached content with the guardband by multiplying Equation (Equation 7) by *G* as Equation (Equation 11). It is calculated by Equation (Equation 11) and is shown in (d) of Figure 3 and Figure 4.
(11)$$C_{CG,i}=\frac{\left(100+G\right)}{100}\times \int_{in}^{out}\frac{r_{w}}{V_{w}\left(w\right)}dw$$

Determining the amount of the precached content using Equation (Equation 11) can solve the problem that occurs in the situation when *V*${}_{Cur}$ is faster than the speed value used in the calculation. On the other hand, in the case of $V_{Cur}<V_{Avg}$, Equation (Equation 11) is derived from Equation (Equation 10) and shown in (b) of Figure 3. The equation for calculating the amount of the precached content that can be provided by considering the guardband is calculated in Equation (Equation 11) and is shown as (d) in Figure 3.

Determining the amount of the precached content by the finally calculated value from Equation (Equation 11) solves the problem that occurs when *V*${}_{Cur}$ is slower than the value used in the calculation. When the requester vehicle stays longer than the expected time in the communication range of the RSU, it can reduce the delay caused by no operation even though it can download more content. In addition, the precaching hit ratio for the amount of the precached content can be increased. If the amount of the content is requested and received from the backhaul at an additional time, the amount of the content received from the backhaul and the amount of the lost content can be reduced. As a result, the impact on the future predicted precached amount of content is reduced, and the amount of the content received in advance in the next RSU is reduced, and thus additional precaching traffic is reduced.

As a result, through the proposed scheme, the amount of the precached content can be downloaded by the requester vehicle. Furthermore, the requester vehicle can be guaranteed for the downloading of the whole content using a guardband. It increases the precaching hit ratio for the amount of the precached content. Also, the traffic and delay are decreased by reducing requests for the amount of the precached content that is not prepared by the next RSU through backhaul links.

## 5. Performance Evaluation

In this section, we compare the performance of the proposed scheme (ACPS) with those of two previous schemes, a current speed scheme [31] and an average speed scheme [30]. We first describe our simulation model and performance evaluation metrics. We next evaluate the performance of the proposed scheme and those of the two previous schemes through simulation results.

### 5.1. Simulation Environment

We compare the performances of the current speed scheme and the average speed scheme with that of the proposed scheme through simulations. The current speed scheme does not consider the speed change of a requester vehicle for calculating the amount of the precached content in the next RSU. On the other hand, the average speed scheme considers when a requester vehicle enters into the next RSU, its speed is equal to the average speed of all vehicles within the communication range of the next RSU. As a result, both the current speed scheme and the average speed scheme use constant speeds. However, the proposed scheme considers that a requester vehicle changes its speed from the current speed to the average speed. To predict this speed change, the speed of the requester vehicle is adaptively changed from the current speed to the average speed by the optimized adaptive value *a*. Thus, the proposed scheme uses the predictive speed of the requester vehicle for calculating the amount of the precached content in the next RSU.

We have implemented the proposed scheme, the current speed scheme, and the average speed scheme in the NS-3 network simulator [45] for comparing their performances. In the NS-3 network simulator, a discrete event simulation models a system in such a way that changes to its state occurrence at discrete points in the simulation time. Table 3 shows the general parameters used in our simulations. The size of our simulated network field is an area of 5000 m × 5000 m which has 25 intersections in urban environments. Each intersection has 1 RSU. Each RSU has a cache storage of 1 GB and its communication coverage is 250 m. The distance between two neighbor RSUs is 1000 m. We set 100 vehicles to move on roads in the simulated network. Each vehicle has 6Mbps communication rate [46,47,48]. For the mobility of vehicles, we apply the Routes Mobility Model [44] which is generally used in NS-3 reflecting real urban environments using the API key from Google Maps Platform for the real speed of the vehicles. The trace from the Routes Mobility Model records the GPS coordinates for 100 vehicles in Brooklyn for more than 1 day with a granularity of one minute. To improve the accuracy of our simulations, we increase the granularity to ten seconds by linear interpolation. Every vehicle has a moving speed of an average 60 Km/h and its speed changes between 20 Km/h and 100 Km/h. Every vehicle has a communication coverage of 100 m and uses the 802.11p (WAVE) [49] protocol as the MAC protocol with a header size of 70 bytes. For the propagation delay and the propagation loss models [50], our simulations use the Constant Speed Propagation Delay Model [51,52] and the Nakagami Propagation Loss Model [53,54], respectively. We set the size of the requested content from 150 to 400 (MB). Each simulation result was conducted over 1000 times with a 95% Confidence Interval (CI).

To evaluate the proposed scheme, we compare its performance with those of the current speed scheme and the average speed scheme in terms of two metrics, the content download delay and the backhaul traffic overhead.

The content download delay is defined as the elapsed time from when the requester vehicle requests an intended content to when the content is fully downloaded by the vehicle.The backhaul traffic overhead is defined as the amount of the precached content that remains in the RSU because the requester vehicle cannot fully download the precached content.

### 5.2. Simulation Results for the Adaptive Value and the Current Speed

Figure 5a shows the content download delay for different adaptive values. The current speed scheme and the average speed scheme use the current speed of the requester vehicle and the average speed of vehicles within the next RSU to calculate the amount of the precached content, respectively. They have the constant content download delay because the current and the average speed are constant values by not considering an adaptive value. On the other hand, the proposed scheme uses the predictive speed of the requester vehicle by adjusting from the current speed to the average speed with an adaptive value. When the current speeds are different (i.e., the current speed n is 30, 60, 100 km/h), the proposed scheme has changes of the content download delay because the requester vehicle changes its speed according to the adaptive value. Thus, the proposed scheme has the optimal value for each current speed.

The proposed scheme with the current speed (30 km/h) has the largest content download delay in the small adaptive value. However, as the adaptive value increases, the content download delay decreases. After the adaptive value is over 20, the content download delay becomes constant. The proposed scheme with the current speed (60 km/h) has a virtually constant content download delay because the requester vehicle’s current speed is the same as the average speed (60 km/h). However, since the proposed scheme considers the adaptive value, it has a lower content download delay than the average speed scheme. The proposed scheme with the current speed (100 km/h) has the lowest content download delay. As the requester vehicle passes the RSU with high speed, the travel time from the current RSU to the next RSU decreases. For each of the current speeds (30 km/h, 60 km/h, and 100 km/h) in the proposed scheme, there is an adaptive value with a parabolic curve for providing the least delay.

Figure 5b shows the normalized content download delay for different adaptive values. In the normalized graph, the proposed scheme has the lowest delay according to the current speed of the requester vehicle. When the current speed is slower than the average speed, the content download delay is the smallest value at the large adaptive values. On the other hand, as the current speed of the requester vehicle is faster than the average speed, the content download delay is lower at smaller adaptive values.

Figure 6a shows the backhaul traffic overhead for different adaptive values. The current speed scheme has a constant and highest backhaul traffic overhead because it only considers the current speed of the requester vehicle, which is a constant value as the speed for calculating the amount of the precached content. The average speed scheme also has the constant backhaul traffic because it uses the average speed of the vehicles in the next RSU, which is also a constant value. The current speed scheme has a larger backhaul traffic overhead than the average speed scheme because the current speed is more different from the practical speed of the requester vehicle than the average speed. On the other hand, the proposed scheme has changes of the backhaul traffic overhead for different current speeds (30, 60, 100 km/h) because it uses the predictive speed of the requester vehicle by using an adaptive value. The proposed scheme with the current speed (30 km/h) has a larger backhaul traffic overhead than the proposed scheme with other current speeds. However, the backhaul traffic overhead gradually decreases. The proposed scheme with the current speed (60 km/h) is little affected by the adaptive values. In other words, the speed of the requester vehicle is little changed, and the backhaul traffic overhead is almost constant. The proposed scheme with the current speed (100 km/h) has a lot of fluctuations because the speed of the requester vehicle is largely affected by the adaptive value. When the adaptive value is under 12, the backhaul traffic overhead gradually decreases and becomes the lowest value at 12.

Figure 6b shows the normalized backhaul traffic overhead for different adaptive values. In the normalized graph, the proposed scheme has an adaptive value with the smallest backhaul traffic overhead according to the current speed of the requester vehicle. The current speed slower than the average speed results in the smallest backhaul traffic overhead at a large adaptive value. Also, as the current of the requester vehicle is faster than the average speed, the backhaul traffic overhead is lower at smaller adaptive values.

### 5.3. Simulation Results for the Current Speed and the Guardband

Figure 7a shows the content download delay for different current speeds of the requester vehicle when the optimal adaptive value is applied for each current speed. The current speed scheme has the highest content download delay in all of the current speeds because it has big differences between the current and the practical speeds of the requester vehicle. The content download delay of the proposed scheme is similar to that of the average speed scheme (60 km/h). On the other hand, when current speeds are higher than the average speed (60 km/h), all of the schemes decrease the content download delays and have similar content download delays. As the current speed is increased, the proposed scheme has the lowest content download delay because it adjusts the speed of the requester vehicle according to the adaptive value.

Figure 7b shows the content download delay of the proposed scheme for different current speeds when it considers different guardbands. As the current speed increases, the content download delay of the proposed scheme decreases because the time that the requester vehicle cannot download the content is decreased due to the fact that its travel time from the current RSU to the next RSU is decreased. Additionally, if the amount of the guardband is added more, the delay caused by the difference between the amount of the downloaded content and the amount of the precached content is decreased because the guardband covers the amount of the precached content that is not downloaded by the error of the speed prediction.

Figure 8a shows the backhaul traffic overhead for different current speeds of a requester vehicle when the optimal adaptive value is applied for each current speed. Since the proposed scheme properly adjusts the current speed to the average speed by applying the optimal adaptive value, the backhaul traffic overhead of the proposed scheme is lower than the current and the average speed schemes. Additionally, when the current speed of the requester vehicle is faster than the average speed, the requester vehicle cannot fully download the precached content and travels to the next RSU because the staying time in the communication coverage of the RSU is decreased. Furthermore, since the next RSU does not have the amount of the precached content in the previous RSU, the next RSU should request the content to the content server through the backhaul links. As a result, it increases the backhaul traffic overhead.

Figure 8b shows the backhaul traffic overhead of the proposed scheme for different current speeds when it considers different guardbands. At the current speed (60 km/h), the proposed scheme only uses the smallest guardband because it adjusts the speed of the requester vehicle little. Therefore, it has the highest backhaul traffic overhead. As the proposed scheme uses more guardband, the probability that the requester vehicle fully downloads the precached content is increased but it causes the increase of the backhaul traffic overhead.

### 5.4. Simulation Results for the Guardband

First, we compare the proposed scheme of different current speeds with the current speed scheme and the average speed scheme for different guardbands in Figure 9. In the proposed scheme, each current speed has the optimal adaptive value.

Figure 9a shows the content download delay for different guardbands. In the proposed scheme, if the requester vehicle remains in the next RSU for a longer time than the expected time, the amount of the precached content is insufficient to be downloaded by the requester vehicle in the communication coverage of the RSU. Thus, the proposed scheme adds the guardband to cover the insufficient amount of the precached content. When the guardband is increased, the proposed scheme decreases the content download delay because it adjusts the speed of the requester vehicle with the adaptive value. On the other hand, the current and the average speed schemes have constant content download delays because they do not consider the guardband.

Figure 9b shows the backhaul traffic overhead for different guardbands. When the amount of the precached content is smaller than the amount of the downloaded content, the guardband added in the precached content is used to be downloaded by the requester vehicle. In the situation that the current speed is equal to the average speed, the guardband is not used to cover the amount of the precached content and thus becomes the backhaul traffic overhead. The proposed scheme with the current speed of 60 km/h has more backhaul traffic overhead than the proposed scheme with the current speeds of 30 km/h and 100 km/h. However, as more guardband is added, the requester vehicle does not download a large part of the guardband and leaves the RSU. As a result, as the guardband increases, the backhaul overhead traffic is increased. On the other hand, the current and the average speed schemes have constant backhaul traffic overheads by not considering the guardband.

Next, we compare the proposed scheme of different adaptive values with the current speed scheme and the average speed scheme for different guardbands in Figure 10. To show the obvious difference between schemes, we set the current speed as 80 km/h, and the optimal adaptive value of *a* is 6.

Figure 10a shows the content download delay for the amount of the guardband when the adaptive values of the proposed scheme are different. When the value of *a* is 0, the requester vehicle does not change its speed. The proposed scheme (*a* = 0) without the guardband has the equal performance to that of the current speed scheme. When the value of *a* is 35, the requester vehicle reduces its speed rapidly to the average speed. Therefore, the performance of the proposed scheme (*a* = 35) without the guardband is similar to that of the average speed scheme. However, as the amount of the guardband is increased, the content download delay is decreased because the guardband adjusts the prediction errors. As a result, calculating the optimal adaptive value *a* is most important, and then the prediction errors are covered by the guardband.

Figure 10b shows the backhaul traffic overhead for the amount of the guardband when the adaptive values of the proposed scheme are different. If the difference between the predictive value of *a* and the optimal value of *a* is large where *a* is 0, the backhaul traffic overhead is largest because the requester vehicle does not change its speed. As the amount of the guardband is larger, the prediction errors are covered larger by the guardband. As a result, the backhaul traffic overhead is decreased. However, if the difference between the value of *a* and the optimal value of *a* is small, the backhaul traffic overhead is increased because the size of the prediction errors covered by the guardband is small. Thus, to minimize the backhaul traffic overhead, it needs to use a suitable amount of the guardband.

### 5.5. Simulation Results for the Current Speed and the Size of the Requested Content

Figure 11a shows the content download delay for different current speeds of a requester vehicle. The current and average speed schemes do not have the performance improvement after the current speed of 60 km/h because they have errors in predicting the amount of the precached content and thus need the additional time to request for the content to the content server to compensate for the errors. Moreover, since the current speed scheme has larger errors, it has a longer content download delay than the average speed scheme. On the other hand, since the guardband of the proposed scheme covers errors in predicting the amount of the precached content, the proposed scheme with a larger guardband has better performance than the proposed scheme with a smaller guardband.

Figure 11b shows the backhaul traffic overhead for different current speeds of the requester vehicle. In the proposed schemes, if the difference between the current speed and the average speed increases, the precached content added by the guardband is more downloaded by the requester vehicle and thus the backhaul traffic overhead is reduced. When the requester vehicle enters the RSU with an average speed of 60 km/h, the backhaul traffic overhead increases because most of the added guardband is not downloaded. When the current speed scheme has larger differences between the current speed and the average speed, it has larger errors in predicting the amount of the precached content and thus generates more backhaul traffic overheads. The average speed scheme reduces the backhaul traffic overhead when the current speed is lower than the average speed (60 km/h). However, it increases the backhaul traffic overhead when the current speed is higher than the average speed.

Figure 12a shows the content download delay for different sizes of requested contents. As the size of the requested content increases, all schemes increase the content download delay because the requester vehicle needs more time to download the content and passes more RSUs to finish the content downloading. Since the current speed scheme has larger errors than the average speed scheme in calculating the amount of the precached content due to more difference between the current speed and the practical speed of the requested vehicle, it has higher content download delay than the average speed scheme. On the other hand, the proposed scheme has lower content download delay by using an additional guardband because guardband reduces the time that the requester vehicle can download the precached content. Thus, the proposed scheme with G = 1.5 has better performance than the proposed scheme with G = 0 and G = 0.8.

Figure 12b shows the backhaul traffic overhead for different sizes of the requested content. In all schemes, as the size of the requested content increases, they have more backhaul traffic overhead to precache larger requested contents in the next RSUs. Since the proposed scheme appropriately calculates the amount of the precached content using the optimized value, it has a lower backhaul traffic overhead than both the average speed scheme and the current speed scheme. Since the current speed scheme has larger errors than the average speed scheme in calculating the amount of the precached content, the current speed scheme has a higher backhaul traffic overhead than the average speed scheme. On the other hand, the proposed scheme causes more backhaul traffic overhead to precache more guardband for bigger guardband. Thus, the proposed scheme with G = 1.5 has more backhaul traffic overhead than the proposed scheme with G = 0 and G = 0.8.

## 6. Conclusions

As the size of content such as multimedia data becomes large, it might be difficult for a vehicle to download the whole of content from a single RSU. To address this issue, many studies exploit precaching of content in each of the next RSUs on the trajectory of the vehicle. For this, previous precaching schemes use the current speed of the vehicle requesting the content or the average speed of vehicles in each RSU to calculate the downloadable amount of the content in the RSU. However, since they do not appropriately reflect the practical speed of the requester vehicle in the RSU, they could not calculate precisely the downloadable amount of the content. Thus, we propose an adaptive content precaching scheme (ACPS) that correctly estimates the predictive speed of a requester vehicle to reflect its practical speed and calculates the downloadable amount of an intended content through using its predictive speed. Moreover, to guarantee fully downloading the amount of the precached content, ACPS adds a guardband area to the downloadable amount to compensate for the difference between the practical and the predictive speeds.

We conducted extensive simulations in various environments to verify the performance of ACPS. In many circumstances, ACPS shows better performance than the current and the average speed schemes in terms of the content download delay and the backhaul traffic overhead. Only, ACPS shows lower performances than the current or average speed schemes when the current speed of the requester vehicle is 30 km/h or 100 km/h until the requester vehicle increases/decreases its speed to 60 km/h. However, as the speed of the requester vehicle is converged to 60 km/h, ACPS mostly achieves better performance than the current or average speed schemes.

In this paper, we consider the general urban road condition. However, there are more different road conditions in the real world, and it causes huge complexity to calculate and precache the downloadable amount of the content. Thus, in the future work, we need to adopt the machine-learning to simplify the complexity of calculation and find the optimal value *a* for the different scenarios. Moreover, since RSUs generally have limited storage resources, they can store only the restricted amount of contents for precaching. Thus, we also need to calculate the optimal amount of the precached content while considering the limited storage resources of RSUs.

## Figures and Tables

**Figure 1 sensors-21-05376-f001:**
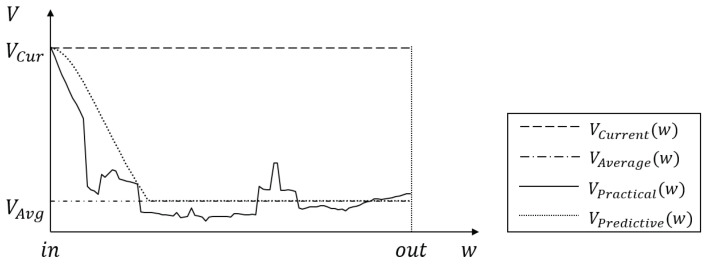
The speed of the requester vehicle according to its locations (w) within the communication range (between the arrival point in and the departure point out) of the next RSU when different speeds are used for calculating the amount of the precached content.

**Figure 2 sensors-21-05376-f002:**
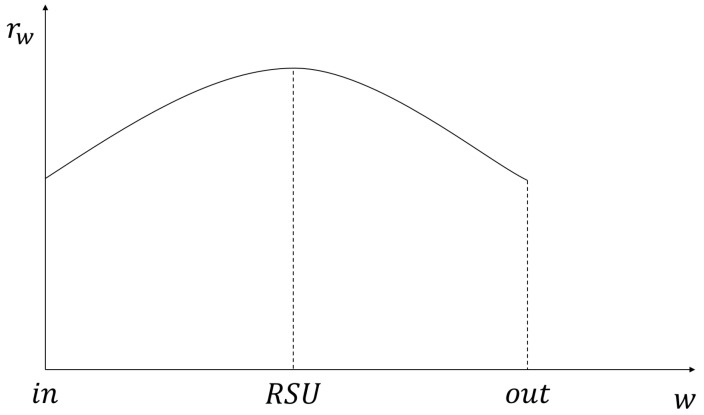
The value of *r${}_{w}$* within the coverage of an RSU.

**Figure 3 sensors-21-05376-f003:**
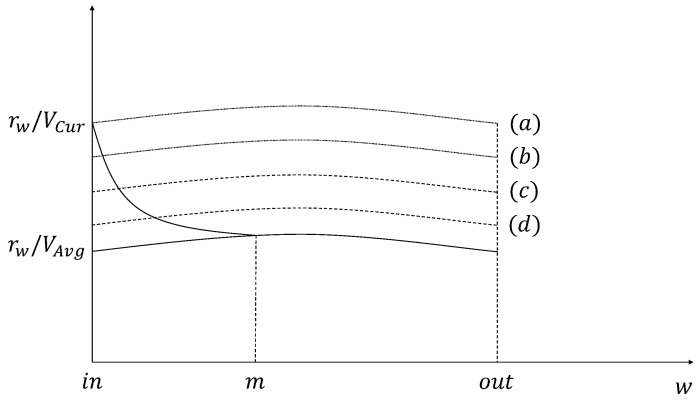
The amount of the precached content when $V_{Avg}$ is faster than $V_{Cur}$.

**Figure 4 sensors-21-05376-f004:**
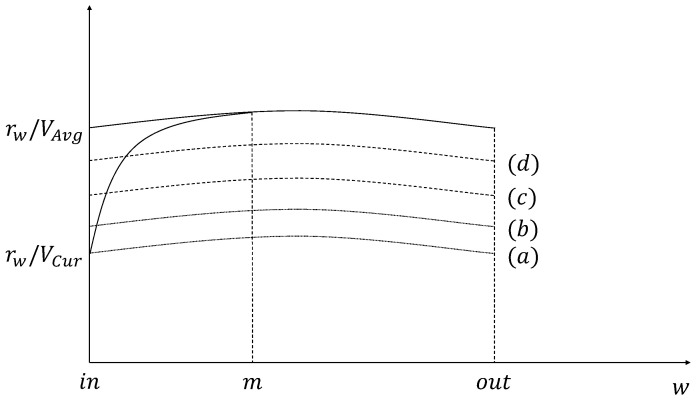
The amount of the precached content when *V*${}_{Avg}$ is slower than *V*${}_{Cur}$.

**Figure 5 sensors-21-05376-f005:**
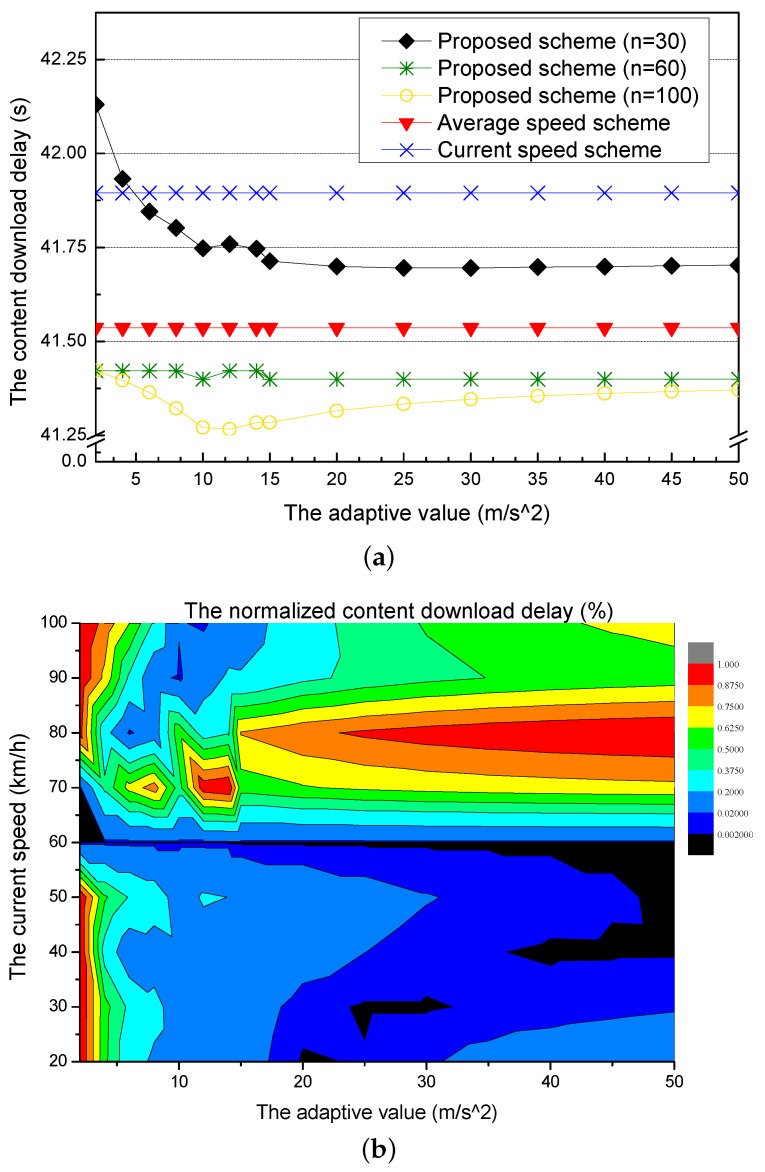
Simulation results for different adaptive values: (**a**) the content download delay and (**b**) the normalized content download delay.

**Figure 6 sensors-21-05376-f006:**
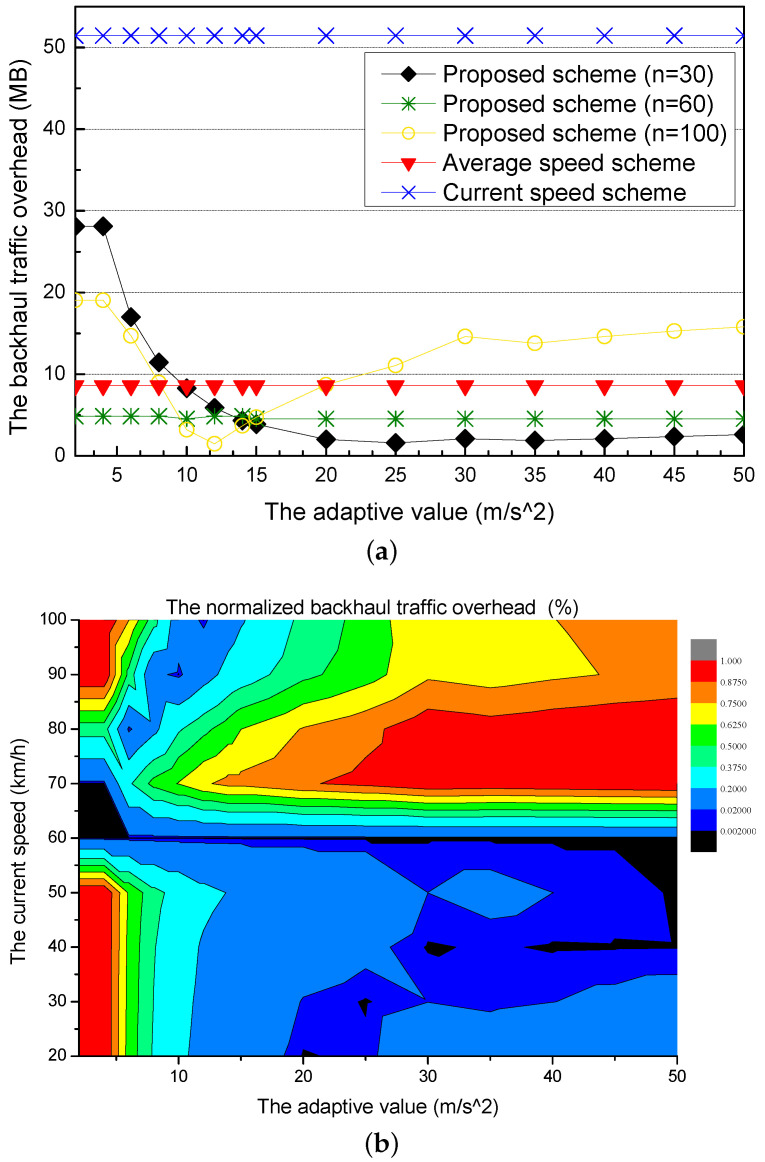
Simulation results for different adaptive values: (**a**) the backhaul traffic overhead and (**b**) the normalized backhaul traffic overhead.

**Figure 7 sensors-21-05376-f007:**
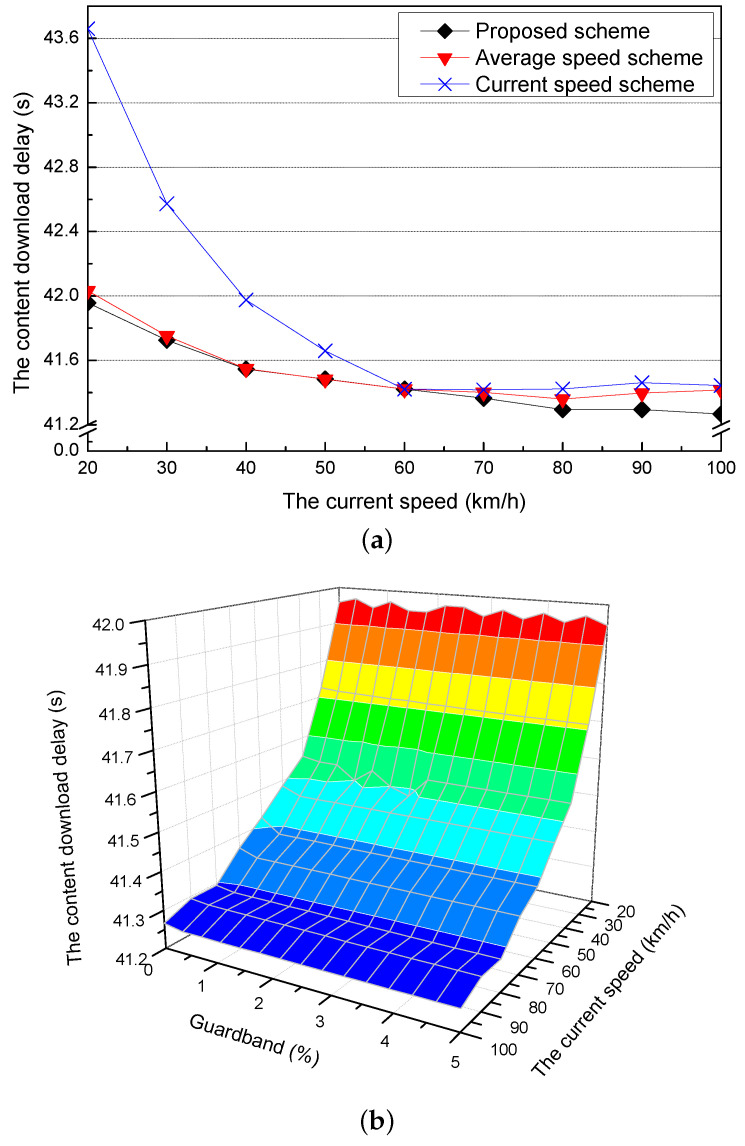
Simulation results for the different current speeds: (**a**) the content download delay and (**b**) the content download delay for different guardbands.

**Figure 8 sensors-21-05376-f008:**
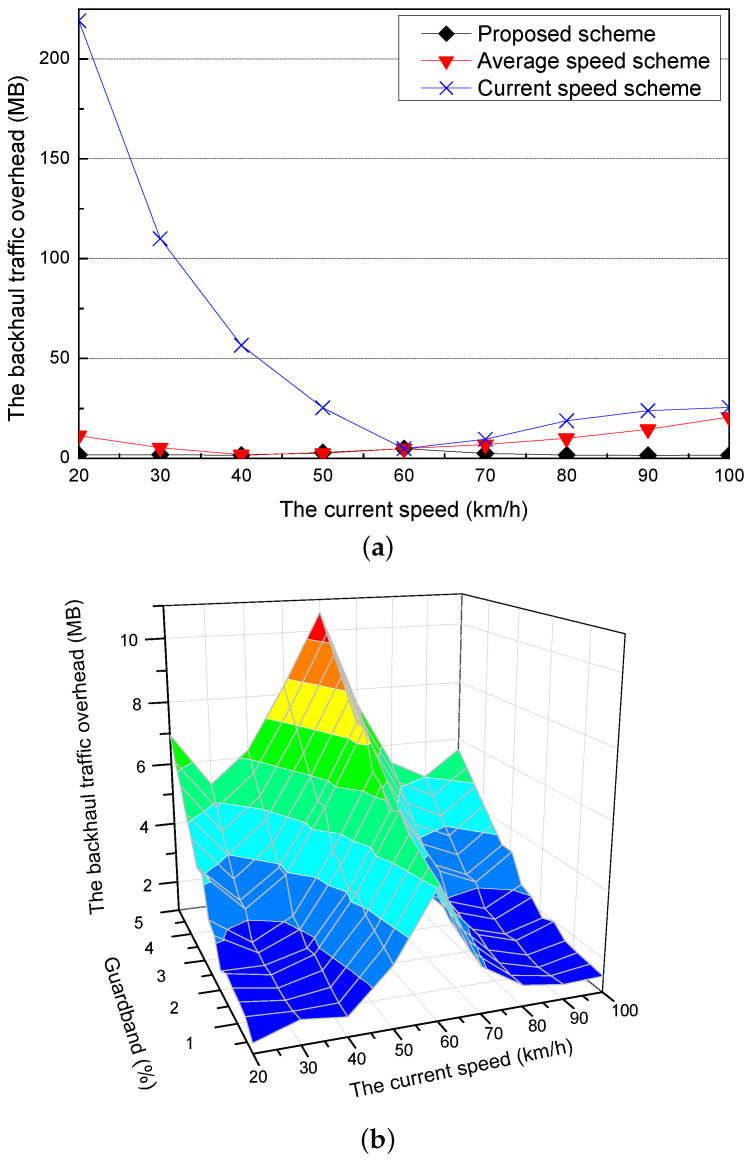
Simulation results for the different current speeds: (**a**) the backhaul traffic overhead and (**b**) the backhaul traffic overhead for different guardbands.

**Figure 9 sensors-21-05376-f009:**
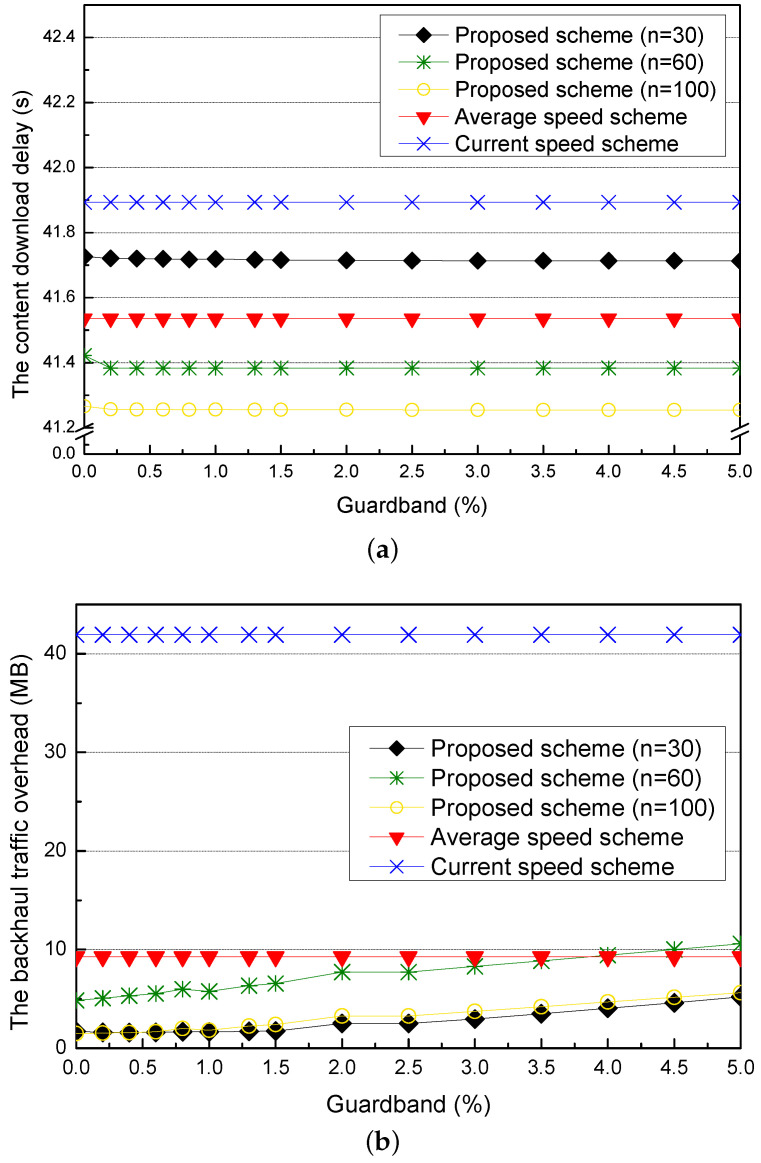
Simulation results for different guardbands: (**a**) the content download delay and (**b**) the backhaul traffic overhead.

**Figure 10 sensors-21-05376-f010:**
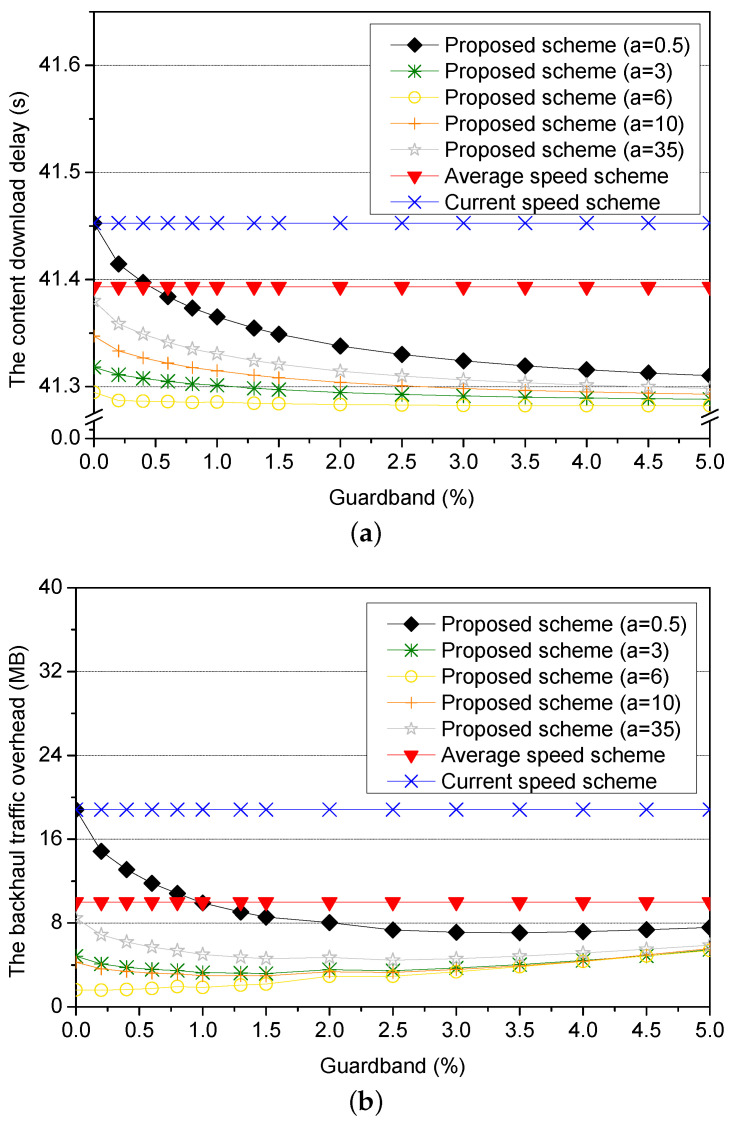
Simulation results for different sizes of the guardband: (**a**) the content download delay and (**b**) the backhaul traffic overhead.

**Figure 11 sensors-21-05376-f011:**
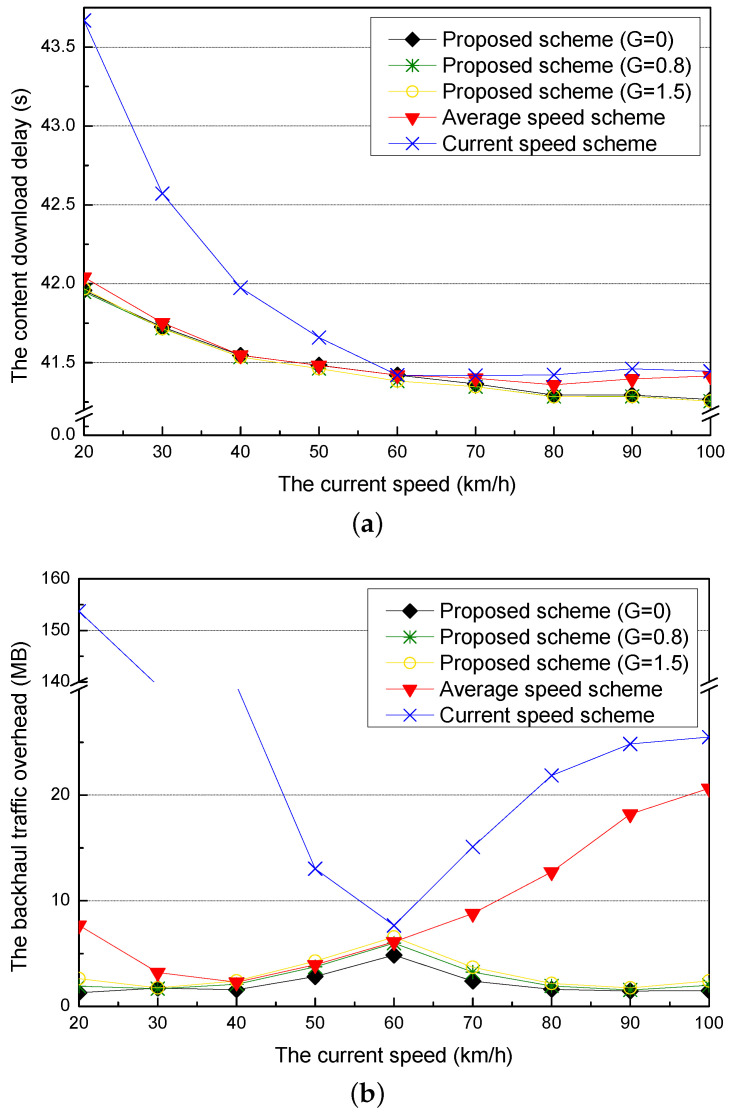
Simulation results for different current speeds: (**a**) the content download delay and (**b**) the backhaul traffic overhead.

**Figure 12 sensors-21-05376-f012:**
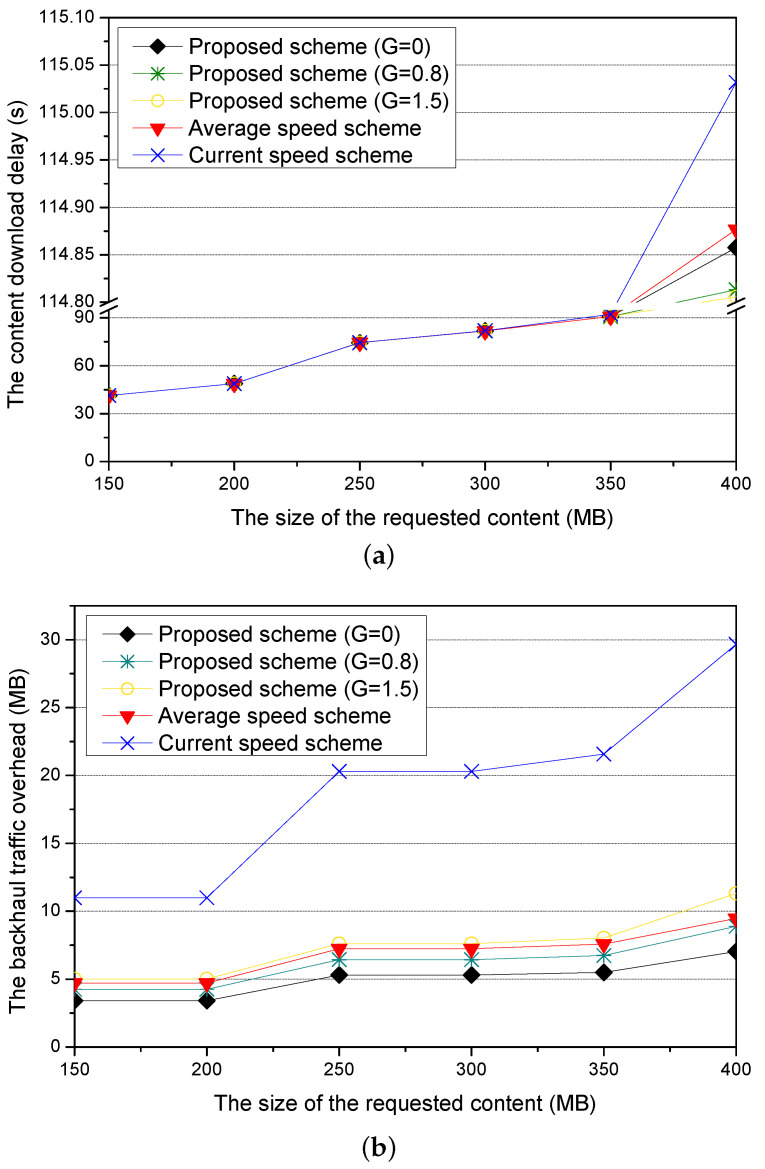
Simulation results for different sizes of the requested content: (**a**) the content download delay and (**b**) the backhaul traffic overhead.

**Table 1 sensors-21-05376-t001:** The summary of the related works on precaching in CCVNs.

Reference	Caching Decision Factors	Mobility Supporting	Cached Amount	Performance Parameters	Method
[17]	mobility of a train	O	Partial	QoE	mathematical calculation
[18]	mobility probability to Hot Regions, popularity	X	Full	success ratio, access delay	mathematical calculation
[19]	backhaul link, wireless resource	X	Full	delay, throughput, profit	optimization, strategy-proof auction mechanism
[20]	mobility, popularity	O	Partial	communication cost	optimization
[21]	mobility prediction, centrality, load degree, popularity	O	Full	reliability, delay	mathematical calculation
[22]	social attributes, trajectory prediction	O	Partial	success ratio, access delay	weight function
[23]	popularity, content size	X	Full	backhaul link cost	Deep Reinforcement Learning
[24]	request frequency, popularity based on Zipf’s law	O	Full	QoS	long short-term memory, Q-learning

**Table 2 sensors-21-05376-t002:** The summary of the related works on precaching in CCVNs.

Reference	Caching Decision Factors	Cached Amount	Performance Parameters	Method	Speed Consideration
[26]	Entropy of mobility probability	Full	latency, server load, cache redundancy	mathematical calculation	
[25]	mobility prediction, popularity based on frequency	Full	latency, traffic	mathematical calculation	
[36]	mobility	Full	QoS, cellular cost	algorithm	
[37]	position	Full	download performance	forwarding	
[38]	position	Full	cache utilization, resolved request ratio	address table	
[39]	popularity, mobility	Full	latency, network traffic	federated deep learning	
[28]	position, velocity, request frequency	Partial	cache utilization, one-hop ratio, resolved request ratio	mathematical calculation	current speed
[27]	speed-density relationship	Partial	download volume, throughput	mathematical calculation	current speed
[29]	trajectory, speed, direction of a vehicle	Partial	latency	optimization	current speed
[40]	trajectory, speed, direction of a vehicle	Partial	network performance, user experience	mathematical calculation	current speed
[31]	mobility, popularity based on rating, request frequency	Partial	network load and QoE	optimization	current speed
[41]	probability of trajectory, entropy	Partial	hit ratio and delay	mathematical calculation	current speed
[30]	transition probability, popularity based on frequency	Partial	hit ratio	mathematical calculation	average speed

**Table 3 sensors-21-05376-t003:** Simulation Parameters.

Parameter	Values
The network size	5000 × 5000 (m${}^{2}$)
The number of RSUs	25
The size of an RSU cache storage	1 GB
The communication rate	6 Mbps (max 54 Mbps)
The communication coverage of RSUs	250 m
The distance between RSUs	1000 m
The average speed within RSUs	60 km/h
The mobility model	Routes Mobility Model
The number of vehicles	100
The communication coverage of vehicles	100 m
The speed of vehicles	20 to 100 (km/h)
The coding and modulation	QPSK of OFDM
The propagation delay	Constant Speed Propagation Delay Model
The propagation loss	Nakagami Propagation Loss Model
MAC protocol	802.11p (WAVE)
The size of the requested content	150 to 400 (MB)

## Data Availability

Not applicable.

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
