# Peer review of "Adaptive Content Precaching Scheme Based on the Predictive Speed of Vehicles in Content-Centric Vehicular Networks"

_sensors, 2021, doi:10.3390/s21165376_

Round 1

Reviewer 1 Report

The authors propose an adaptive solution for a content precaching scheme based on information on the predictive speed of vehicles. The idea sounds interesting, scientific, and applied to content-centric vehicular networks context. The Main highlighted points are described following:

1) Abstract Section focused only on the motivation of the envisaged problem. Literature gaps and a brief introduction about the innovation of the proposal and results are missing. This section has the potential to be shortened by 20% or more. 

2) Introduction Section presents the target problem under study and describes a bit of the gap in the literature. However, despite the cited references were interesting, the envisaged topic is a hot area nowadays where recent brand-new contributions are found in the literature. On the other hand, references are outdated, and it is missing works from 2020 and 2021. Therefore, a big-picture about the target problem should be introduced to clarify where the contributions are placed faced the other found in the literature. Clarification of contributions is significant to highlight the innovations of the proposal. A itemize bullets with all contributions should be explicitly added too.

3) Section 2 is attractive. The authors split that section in a very didact way. However, references are a bit outdated. Recent contributions from 2020 and 2021 (indeed) are found in the literature. At least 30% to 40% additional references should be introduced to refresh the discussion of Section 2. Additionally, authors should present a Table summarizing and comparing all literature contributions and the proposal at the end of Section 2. 

4) Section 3 is very lengthy, and authors must reduce it. For the sake of understanding, a figure could overview the problem, and the text will follow with a didact description. One page or 1,5 pages are enough here. 

5) The authors could present theorems and proofs in an appendix. Reorganize them is attractive to the text gain readability. 

6) Models presented in Section 4 are very similar to the new literature contributions. The introduction of brand-new references is fundamental to clarify contributions. 

7) Results presented in Section 5 are promising; however, the reader can identify several flaws. First of all, for reproducibility, it is recommended the authors provide the source code used in the paper to make results reproducible. A public repository (e.g gitlab, github or similar) will solve this challenge. Secondly, evaluation scenarios need to be defined at the beginning of this section. Thirdly, and the most important, results are all linear. Therefore, it would be expected some of the stochastic behavior of them. How could the author explain this a little bit more?

8) Indication of future studies is missing in the Conclusion Section. It would be expected a final discussion about the results. It is missing too. 

More detailed comments:

1) It is not clear in the simulation experiment the methodology adopt to extract results. a) How many experiments were conducted? b) Confident interval is missing (this is fundamental because results were only linear); c) presence of noise in the communication channel was not mentioned and this can impact the results; 
2) The range of axis y of figures could be resized to (0,max values).
3) As speed is a crucial variable under study, indicate urban and highway scenarios will make more sense to discuss results. In this case, figures 9 and 10 should be rebuild. 
4) Considering a more adequate number of experiments (that guarantee an interval of confidence, eg. 0.95 or more), the results in Figures 9 and 10 could indicate a curve softer. 
5) Why authors choose NS-3 and not another specific vehicular network simulator?

Reviewer 2 Report

In this paper, authors present a we propose an adaptive content precaching scheme (ACPS) that correctly estimates the predictive speed of a requester vehicle to reflect its practical speed and calculates the downloadable amount of an intended content using its predictive speed. The APCS proposed scheme adjust the predictive speed of the requester vehicle in a next RSU to the average speed of the vehicles in that RSU starting from its current speed.

The paper is well written, and the introduction section provides a good perspective of the problem. Authors also include a good related work section with recent references.

The proposal method is well described and the algorithm provide better results than other state-of-the-art proposals.

The paper has an extended experiment section. May be it could be shortened.

However, in my opinion the conclusion section does not reflect any information about the presented results. For example, when, according to the speed(n=30,60,..), is preferable to use current speed or average speed instead of the proposed scheme, respect to delay, backhaul, etc?

Reviewer 3 Report

This paper mainly proposed an adaptive content precaching scheme that correctly estimates the predictive speed of a requester vehicle to reflect its practical speed and calculate the downloadable amount of an intended content through using its predictive speed. The topic was interesting; however, the paper should be improved from the following points,

  • There are some typos and grammar mistakes in the paper which make some places of the paper difficult to read. For example, “a vehicle a vehicle has difficulty downloading the whole amount of content within the coverage of the RSU where it is located in [10]”
  • For Figure 3, the meanings of (a) and (c) were explained in the paper, how about (b) and (d)
  • “we compare the performance of the proposed scheme (ACPS) with those of two previous schemes, a current speed scheme [18] and an average speed scheme [19]” Please kindly explain why you choose current speed scheme and average speed scheme. Are they the best in the previous studies?
  • “we set the current speed as 80km/h, and the optimal adaptive value of a is 6” Please kindly provide the reason that the optimal adaptive value of a was set as 6, and why do you choose 0, 3, 10, 35 as companions
  • “In the equation (24), G is a constant value between 0 and 1”, for figures (15) and (16), why G was set as 0, 0.8 and 1.5?

Round 2

Reviewer 1 Report

The authors have been implemented a major revision in the paper considering all suggestions mentioned in the first revision round. It is possible to note the overall quality of the paper was improved and gaps related to clarification of the proposal were solved. Thus, the reviewer suggest accept the paper in the present form. 

Reviewer 3 Report

Thank you for your response file.